# Long-Term Clinical Outcome and Predictive Factors for Relapse after Radiation Therapy in 145 Patients with Stage I Gastric B-Cell Lymphoma of Mucosa-Associated Lymphoid Tissue Type

**DOI:** 10.3390/cancers13020169

**Published:** 2021-01-06

**Authors:** Heerim Nam, Do Hoon Lim, Jae J. Kim, Jun Haeng Lee, Byung-Hoon Min, Hyuk Lee

**Affiliations:** 1Department of Radiation Oncology, Kangbuk Samsung Hospital, Sungkyunkwan University School of Medicine, Seoul 03181, Korea; heerim.nam@gmail.com; 2Department of Radiation Oncology, Samsung Medical Center, Sungkyunkwan University School of Medicine, Seoul 06351, Korea; 3Department of Medicine, Samsung Medical Center, Sungkyunkwan University School of Medicine, Seoul 06351, Korea; jjkim@skku.edu (J.J.K.); stomachlee@gmail.com (J.H.L.); jason1080.min@samsung.com (B.-H.M.); leehyuk@skku.edu (H.L.)

**Keywords:** gastric MALT lymphoma, radiotherapy, long-term outcome, tumor location

## Abstract

**Simple Summary:**

*Helicobacter pylori*-associated gastric low-grade B-cell lymphoma of mucosa-associated lymphoid tissue type (MALT lymphoma) constitutes >80% of gastric MALT lymphoma. Eradication therapy has been accepted as a standard approach for initial treatment. However, in patients who present without evidence of infection or who fail to respond to eradication therapy, a solid consensus for treatment is not available. Furthermore, few studies have evaluated the predictive factors for response or relapse after radiation therapy (RT) as heterogeneous, relatively small study populations have been treated with RT, and only a small number of events have been reported after treatment. In this study, we report the long-term clinical outcome of stage I gastric MALT lymphoma treated with RT. We also identified that the tumor’s dominant location in the stomach is a predictive factor for relapse after RT.

**Abstract:**

This study aimed to evaluate the clinical outcomes of radiation therapy (RT) for stage I gastric mucosa-associated lymphoid tissue (MALT) lymphoma and find predictive factors for relapse after RT. This retrospective study included 145 patients without a prior history of treatment, except *Helicobacter pylori* eradication therapy, who were irradiated for stage I gastric MALT lymphoma. The gastric body was the most commonly involved location of the dominant lesion (66.9%), and *H. pylori* infection at first diagnosis was detected in 61 (42.1%) patients. The median RT dose was 30 Gy (range, 24–40). Seven patients had an autoimmune disease. All patients except one achieved a complete remission at post-treatment endoscopic biopsy after a median of 2 months (range, 1–36). During the median follow-up at 51 months (range, 2–146), 11 patients experienced relapses: in the stomach (*n* = 5), in a distant site (*n* = 4), and in both (*n* = 2). The five-year overall, local relapse-free, distant relapse-free, and relapse-free survival (RFS) rates were 98.6%, 94.0%, 97.1%, and 92.3%, respectively. In multivariate analysis for RFS, the location of MALT lymphoma other than in the gastric body was significantly associated with an increased risk of relapse (hazard ratio 5.85 (95% CI 1.49–22.9), *p* = 0.011). RT results in favorable clinical outcomes in patients with stage I gastric MALT lymphoma. Tumor location could be a predictive factor for relapse after RT.

## 1. Introduction

Approximately 70% of marginal zone B-cell lymphomas (MZL) are extra-nodal mucosa-associated lymphoid tissue (MALT) and often termed as MALT lymphomas. The most common site for the occurrence of MALT lymphoma is the stomach, which accounts for 50–70% of all MALT lymphomas [1].

MALT lymphoma typically shows an indolent clinical course, and 60–80% of cases are diagnosed at an early, localized stage [2]. The strong association between *Helicobacter pylori* and gastric MALT lymphoma is well studied, and >80% patients with gastric MALT lymphoma have an *H. pylori* infection [3]. *H. pylori* eradication is considered the first-line treatment option and leads to long-term control of lymphoma in >60% cases [4,5].

However, in cases of localized stage I gastric MALT lymphoma that is not associated with *H. pylori* infection or in cases that fail to respond to antibiotic therapy, no solid consensus for the optimal treatment modality exists, and systemic therapy, surgery, the watch-and-wait approach or radiation therapy (RT) are selected according to the treating physician’s preference or expertise [6]. Although clinical guidelines recommend RT as the preferred initial therapy for these cases, RT is underused [6,7].

Studies on gastric MALT lymphoma do not consistently assess patients’ clinical outcomes following RT, mostly due to the small number of patients, inhomogeneous primary treatment, and insufficient follow-up. In this study, we describe the responses to RT and long-term outcomes of stage I gastric MALT lymphoma. We also aim to identify predictive factors for relapse after RT.

## 2. Results

### 2.1. Patient Characteristics

The median follow-up of 145 patients was 51 months (range, 2–146 months). Patient characteristics are summarized in Table 1 and Figure 1. The median patient age was 53 years. In total, 44.8% of patients were female. The gastric body was the most commonly involved location with a dominant lesion, and *H. pylori* infection at first diagnosis was detected in 61 (42.1%) patients. 

Antibiotic treatment for *H. pylori* eradication was administered to 87 (60%) patients, including 26 patients without *H. pylori* infection. Before initiation of RT, *H. pylori* infection was eradicated in 57 patients. All patients had endoscopic evidence of a tumor before RT. Tumor responses to eradication therapy before initiation of RT were responding residual disease (rRD) in 15 patients, and no change (NC) in 51 patients. Fifteen patients had recurrent tumors after attaining complete remission (CR), and six patients had progressed lesions after eradication therapy.

The median time interval from diagnosis to initiation of RT was 16 months (range, 3–43) for rRD, 11 months (range, 3–119) for NC, 20 months (range, 9–56) for recurrent tumor, and 4 months (range, 2–41) for progressive lesions after eradication therapy. Radiation therapy was initiated at a median of one month (range, 1–13) after diagnosis in patients without *H. pylori* infection.

Seven (4.8%) patients had an autoimmune disorder diagnosis, including idiopathic thrombocytopenic purpura (ITP) in two patients, Sjogren’s disease and Hashimoto’s thyroiditis in one patient, ulcerative colitis in one patient, rheumatoid arthritis in one patient, IgA nephropathy with kidney transplantation in one patient, and mixed connective tissue disease in one patient.

### 2.2. RT Response

All patients except one showed CR at the post-treatment endoscopic biopsy after a median of two months (range, 1–36 months). Twenty-eight (28/144, 19.4%) patients did not achieve CR at the first post-treatment endoscopic biopsy. Of these patients, 27 patients showed probable minimal residual disease (pMRD) for a median of five months (range, 4–36 months) before achieving CR. One patient showed rRD for 20 months before achieving CR.

### 2.3. Disease Relapse, Survival, and Predictive Factor for Relapse after RT

The 5-year and 10-year overall survival (OS) rates were 98.6% and 94.1%, respectively (Figure 2). Two patients died of intercurrent disease. The cause of death was colon cancer in one patient and acute myeloid leukemia in the other patient. 

During the median follow-up of 51 months (range, 2–146 months), 11 patients experienced relapses—in the stomach (*n* = 5), in a distant site (*n* = 4), and both (*n* = 2). The 5-year and 10-year LRFS, DRFS, and RFS were 94.0% and 90.2%, 97.1% and 87.6%, and 92.3% and 82.7%, respectively (Figure 2). DR was reported in six patients. One patient had pathologically confirmed nodal large B-cell lymphoma of the neck. The other patients with distant relapse had: bronchus-associated lymphoid tissue lymphoma (two patients), orbital MALT lymphoma (one), colonic MALT lymphoma (one), and rectal MALT lymphoma (one). The details of patients with relapse are described in Table 2 and Table 3. Six of the seven patients with LR attained second CR after observation (three patients), *H. pylori* eradication (two patients), and chemotherapy (one patient). One patient (patient 5) had NC lymphoma during the 58-month follow-up without treatment. 

In both univariate and multivariate analyses for RFS, the dominant location of MALT lymphoma was significantly associated with relapse (Table 4). The 5-year RFS of patients with dominant tumor in the gastric body vs. other locations was 98.9% vs. 81.5% (*p* = 0.006), and the hazard ratio for the risk of relapse was 5.85 (95% confidence interval [CI], 1.49–22.9, *p* = 0.011) for tumor locations other than the gastric body (Figure 3). Age, presence of autoimmune disease, and *H. pylori* eradication therapy before RT were not significant predictors of RFS.

### 2.4. Toxicities and Secondary Malignancies

During RT, some patients experienced mild epigastric soreness and nausea. During follow-up, seven cases of metachronous malignancies were diagnosed. Six cases were diagnosed with solid malignancies, and one case was diagnosed with acute myeloid leukemia. One patient was diagnosed with gastric cancer in the antrum 21 months after RT. Other solid malignancies were lung cancer (two patients) and colon cancer, prostate cancer, and pancreatic cancer (one patient each).

## 3. Discussion

Marginal zone lymphoma of MALT is a low-grade B-cell lymphoma with an indolent clinical course. Two-thirds of MALT lymphomas are extra-nodal, and the most commonly involved site is the stomach [8].

*H. pylori* eradication therapy is a well-established, first-line treatment recommendation in patients with *H. pylori* infection [5]. However, in patients without evidence of infection or who fail to respond to eradication therapy, a firm consensus on treatment is not available. Although clinical guidelines recommend RT as the preferred therapy for early-stage, non-gastric, and *H. pylori*-negative gastric MALT lymphoma [5,9,10], RT is underused [6]. A recent analysis of 22,378 stage I–II marginal zone lymphoma patients in the National Cancer Database showed that RT was significantly associated with an improved OS [6]. In another analysis using a cohort of 347 patients with stage I gastric MALT lymphoma, RT was associated with a lower lymphoma-related death rate (hazard ratio [HR] 0.27, *p* < 0.001) [11].

In this study, we report on the outcomes of 145 patients with stage I gastric MALT lymphoma who were treated with the modern RT technique. We observed CR in all patients, except one, at a median of two months (range, 1–36 months) after RT. The five-year LRFS rate was 94% and toxicity was minimal. There were no differences in treatment outcomes with RT dose (≤30 Gy vs. >30 Gy). Similar to our study, series with long-term follow-up after RT for gastric MALT lymphoma have reported ≥90% local control rates with standard RT doses of 30–36 Gy [12,13,14]. Given the highly favorable outcomes in patients with gastric MALT lymphoma using an RT dose of 30 Gy, studies regarding dose de-escalation are being conducted. One recent study [15] reported treatment outcomes in 32 gastric MALT lymphoma patients, including 11 patients irradiated with a reduced dose of 24 Gy. After a median follow-up of 27.7 months, there was no relapse in the 24 Gy group. In a recent randomized trial comparing RT doses of 25.2 Gy and 36 Gy, no relapse was found in 22 patients after a median follow-up of 79 months [16].

There has been no widely accepted prognostic index for MALT lymphoma. Recently, the International Extranodal Lymphoma Study Group (IELSG) built a prognostic index with three clinical features using data from 401 patients enrolled in a prospective randomized clinical trial [17]. Among the study population, 42.6% of patients had primary gastric MALT lymphoma. Age ≥ 70 years, Ann Arbor stage III or IV, and elevated LDH levels were significant prognostic parameters [17].

Because early-stage gastric MALT lymphomas have excellent OS, predictive factors for relapse might have greater significance in clinical practice. Because >80% of gastric MALT lymphomas are related to *H. pylori* infection, predictive factors for responses to eradication therapy are relatively well studied. Status of t(11;18)/API2-MALT1 [16,18], MIB-1 status [19], age [20], presence of an autoimmune disease [21,22], and gastric location [23,24,25] have been studied as predictive factors for *H. pylori* eradication therapy.

However, limited studies have evaluated the predictive factors for response or relapse after RT as heterogeneous. Relatively small study populations have been treated with RT, and a small number of events have been reported after treatment. In a long-term follow-up study of the IELSG [12], the presence of large B-cell components or an exophytic growth pattern was reported to be a predictive factor of treatment failure after RT in gastric MALT lymphoma. Multifocal disease or a history of previous treatment before RT had no significance in their analysis. In our study, the only significant factor for relapse was that the dominant tumor location was in the stomach. The five-year RFS of patients with the dominant tumor in the gastric body vs. other locations was 98.9% vs. 81.5% (*p* = 0.006), and the hazard ratio for the risk of relapse was 5.85 (95% CI, 1.49–22.9, *p* = 0.011) for a tumor location other than the stomach. Age, presence of an autoimmune disease, and *H. pylori* eradication therapy before RT were not significant predictors of RFS.

Some studies have indicated that the proximal tumor location is predictive of treatment failure after *H. pylori* eradication therapy [23,24,25]. Although gastric MALT lymphoma can be found anywhere in the stomach and is known to be a multifocal disease [26], this tumor is most commonly localized in the distal part of the stomach [27]. This relative predilection for the distal stomach is explained by the highest concentration of *H. pylori* and acquired lymphoid tissue [28]. In a study on treatment outcomes after *H. pylori* eradication for *H. pylori*-infected gastric MALT lymphoma, patients with distal tumors had a higher CR rate (92.5%) than those with proximal tumors (65.5%) [24]. The possibility of autoimmune-related tumors was presented by the authors [24]. In a study by Wöhrer et al. [22], who performed prospective routine clinical and serological assessments of every MALT lymphoma for autoimmune disease, patients with an autoimmune disease had significantly lower response rates to *H. pylori* eradication therapy compared to patients without an autoimmune disease. Only one of 14 patients with an autoimmune disease (7%) responded to antibiotic therapy. 

In our study, seven patients had an autoimmune disease, and six of these patients had *H. pylori* infection at initial diagnosis. However, none of these patients with an autoimmune disease experienced a relapse. We could not find a study reporting tumor localization in the stomach as a predictive or prognostic factor after RT. Considering that the candidates of RT for gastric MALT lymphoma are patients who failed *H. pylori* eradication therapy or are not related to *H. pylori* infection, shared genetic alterations related to treatment failure with *H. pylori* eradication might also contribute to treatment failure after RT. 

Induction of tumor cell apoptosis is part of the RT-induced cell death mechanism. Translocation t(11;18) (q21;q21) is detected in one of four patients with gastric MALT lymphoma [29,30], and API2-MALT1 generated from this translocation induces an apoptosis inhibitor [31]. The presence of this transcript is a well-known predictive factor for treatment failure of *H. pylori* eradication therapy in gastric MALT lymphoma. However, there are only a few studies regarding the significance of t(11;18)/API2-MALT1 in other treatment modalities. Levy et al. have reported that the presence of t(11;18) in gastric MALT lymphoma is predictive of relapse after using oral alkylating agents [32]. 

The reported frequency of t(11;18)/API2-MALT1 is higher in *H. pylori*-negative gastric MALT lymphoma than in *H. pylori*-positive diseases, and >50% cases are known to have this genetic aberration [25,33]. Advanced-stage [33] and multiple organ involvement in MALT lymphoma have also been studied as relevant factors for a higher frequency of t(11;18)/API2-MALT1 [34].

## 4. Materials and Methods

### 4.1. Patient Selection

Patients with gastric MALT lymphoma were referred for RT if they were negative for *H. pylori* infection or failed to respond to 2 weeks of antibiotic therapy for *H. pylori* eradication.

Of the 162 consecutive patients who were irradiated for stage I gastric MALT lymphoma according to the Lugano staging system [35] at Samsung Cancer Center between January 1998 and December 2019, 145 patients were included in this retrospective study after excluding 15 patients treated with chemotherapy or surgery before RT, and 2 patients who were lost to follow-up without post-RT endoscopic evaluation. All patients had biopsy-proven MALT lymphoma. The institutional review board of Samsung Cancer Center approved this retrospective study (IRB No. SMC 2020-11-070).

### 4.2. Staging Workup

We performed staging workup with esophagogastroduodenoscopy (EGD), positron emission tomography (PET), and/or computed tomography (CT). We also evaluated the complete blood count, biochemistry profile, and lactate dehydrogenase (LDH) levels. Bone marrow biopsy was performed at the discretion of the physician. *H. pylori* infection was tested using histology. If the presence of active *H. pylori* infection was not demonstrated, a urea breath test and/or serologic test were performed.

### 4.3. Evaluation and Follow-Up

Endoscopic evaluation with biopsy was performed 1–3 months after completing RT, every 6 months for the first 2–3 years, and yearly thereafter. The Groupe d’Etude des Lymphomes de l’Adulte histological grading system [36] was used for post-treatment response evaluation. Local relapse (LR) was determined as biopsy-proven gastric MALT lymphoma after initial complete remission (CR) or no change (NC) after RT without achieving CR during follow-up. The detection of MALT lymphoma at an entirely new site was classified as distant relapse (DR). Disease transformation into large B-cell lymphoma was also considered a relapse.

### 4.4. Radiation Therapy

Two-dimensional (2D) simulation using fluoroscopy was performed for seven patients who were treated before 2006. We included a 2 cm margin from the outline of the stomach wall to cover the perigastric lymph nodes after the ingestion of a barium suspension. A parallel opposing field was used for these patients. 

Three-dimensional (3D) simulation with or without the four-dimensional (4D) CT was performed in 79 patients, and intensity-modulated radiation therapy (IMRT) with the 4D CT was administered in 59 patients. The clinical target volume (CTV) was the entire stomach plus the perigastric lymph nodes. The planned target volume (PTV) was individualized, considering setup error and stomach movement. In cases where 4D CT was used, the internal target volume (ITV) was defined at every respiratory cycle. The CTV was set as the ITV plus a 1 cm margin, and the PTV was CTV plus a 1 cm margin. The median RT dose was 30 Gy (range, 24–40 Gy), with a daily dose of 2 Gy per fraction. At our institution, the RT dose for stage I gastric MALT lymphoma was reduced to 30 Gy since 2011.

### 4.5. Statistical Analysis

The endpoints were overall survival (OS), LR-free survival (LRFS), DR-free survival (DRFS), and relapse-free survival (RFS). Follow-up began from the initiation of RT for all endpoints. All local relapses were confirmed by biopsy. The Kaplan–Meier method was used to estimate survival outcomes. Univariate survival comparison was performed using the log-rank test. Multivariate analysis was performed for RFS using the Cox proportional hazards model with a stepwise selection of variables. R (version 3.6.3, Vienna, Austria) was used for all statistical analyses, and a *p*-value of <0.05 was considered statistically significant.

## 5. Conclusions

RT results in favorable clinical outcomes in patients with stage I gastric MALT lymphoma. Tumor location could be a predictive factor for relapse after RT. The main limitations of this study were its retrospective design and the relatively small number of events for analysis. Further studies investigating the relevance of dominant tumor localization and underlying genetic and pathophysiological mechanisms for relapse will help tailor the therapeutic and follow-up strategy of patients with early-stage gastric MALT lymphoma.

## Figures and Tables

**Figure 1 cancers-13-00169-f001:**
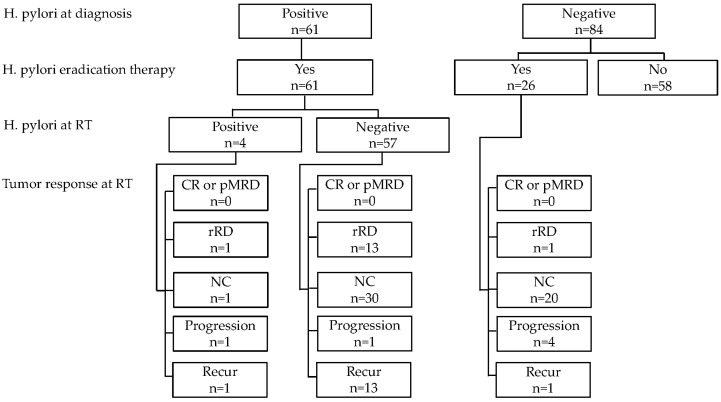
*Helicobacter pylori (H. pylori)* eradication therapy and response before radiation therapy (RT). CR, Complete remission; rRD, Responding residual disease; NC: No change.

**Figure 2 cancers-13-00169-f002:**
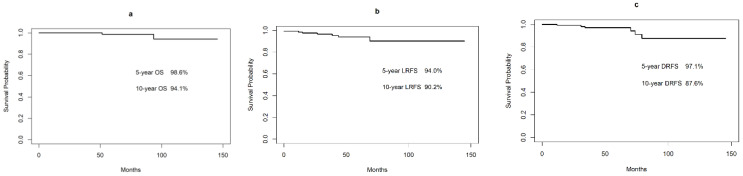
The 5-year and 10-year overall survival (OS) rates were 98.6% and 94.1%, respectively (**a**). During a median follow-up of 51 months, 11 patients experienced relapses. The 5-year and 10-year local relapse-free survival (LRFS) (**b**) and distant relapse-free survival (DRFS) (**c**) rates were 94.0% and 90.2%, and 97.1% and 87.6%, respectively.

**Figure 3 cancers-13-00169-f003:**
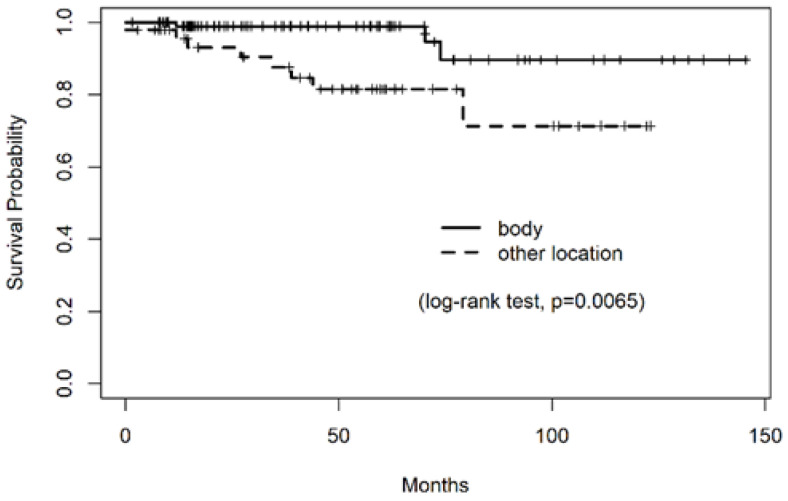
Kaplan–Meier curves for relapse-free survival (RFS) stratified by the dominant location of MALT lymphoma in the stomach. The 5-year RFS of patients with the dominant tumor in the gastric body vs. other locations was 98.9% vs. 81.5% (*p* = 0.006).

**Table 1 cancers-13-00169-t001:** Patient and treatment characteristics.

Characteristics	N = 145
Median age (range)	53 (24–80) years
Sex	
Male	80 (55.2%)
Female	65 (44.8%)
ECOG performance status	
0	125 (86.2%)
1	20 (13.8%)
Location of dominant lesion	
Cardia	0 (0%)
Fundus	26 (17.9%)
Body	97 (66.9%)
Antrum	16 (11.0%)
Diffuse	6 (4.1%)
*Helicobacter Pylori* at diagnosis	
Positive	61 (42.1%)
Negative	84 (57.9%)
Antibiotics	
Yes	87 (60.0%)
No	58 (40.0%)
*Helicobacter Pylori* at RT	
Positive	4 (2.8%)
Negative	141 (97.2%)
Tumor response at RT	
No antibiotics	58 (40.0%)
Complete remission/Probable minimal residual disease	0 (0%)
Responding residual disease	15 (10.3%)
No change	51 (35.2%)
Progression	6 (4.1%)
Recur	15 (10.3%)
Interval to RT, median (range)	
No antibiotics	1 (1–13) months
Responding residual disease	16 (3–43) months
No change	11 (3–119) months
Progression	4 (2–41) months
Recur	20 (9–56) months
RT dose	
Median (range)	30 (24–40) Gy
≤30 Gy	104 (71.7%)
>30 Gy	41 (28.3%)
RT technique	
2D	7 (4.8%)
3D	5 (3.4%)
3D with 4D CT	74 (51.0%)
IMRT with 4D CT	59 (40.7%)

Data are presented as number of patients (%) unless otherwise indicated. Abbreviations: ECOG, Eastern Cooperative Oncology Group; RT, radiation therapy; IMRT, intensity modulated radiation therapy.

**Table 2 cancers-13-00169-t002:** Patients with local relapse.

Patient	Age /Sex	Location	Initial HP Status	HP Eradication	RT Dose	HP at Relapse	Treatment for Relapse	Final Response	LRFS	Distant Relapse, Secondary Malignancy, Autoimmune Disease
1	61/M	Fundus	Positive	Yes	30 Gy	Negative	Chemotherapy+Rituximab	CR	69	Orbital MALT
2	69/M	Body	Negative	No	40 Gy	Negative	observation	CR	12	Neck DLBL
3	47/M	Antrum	Negative	No	36 Gy	Positive	HP irradication	CR	44	AML
4	35/F	Antrum	Negative	No	36 Gy	Positive	HP irradication	CR	39	No
5	63/M	Fundus	Positive	Yes	30 Gy	Negative	observation	NC	0	Pancreatic adenocarcinoma
6	61/F	Antrum	Positive	Yes	36 Gy	Negative	observation	CR	15	No
7	49/F	Diffuse	Negative	No	36 Gy	Negative	observation	CR	27	No

Abbreviations: HP, *Helicobacter pylori*; CR, complete remission; NC, no change; LRFS, Local relapse free interval; MALT, marginal zone lymphoma of mucosa-associated lymphoid tissue; DLBL, diffuse large B-cell lymphoma; AML, acute myeloid leukemia.

**Table 3 cancers-13-00169-t003:** Patients with distant relapse.

Patient	Age /Sex	Location	Initial HP Status	HP Eradication	Relapse	Treatment for Relapse	Status at Last FU	DRFS	Local Relapse, Secondary Malignancy, Autoimmune Disease
1	61/M	Fundus	Positive	Yes	Orbit, MALT	RT	NED	12	Local relapse
2	69/M	Body	Negative	No	Neck node DLBL	Chemotherapy	NED	12	Local relapse
3	72/M	Fundus	Positive	Yes	Lung, BALT	Chemotherapy	NED	79	No
4	57/M	Antrum	Negative	Yes	Colon, MALT	Endoscopic resection	NED	34	No
5	29/F	Body	Positive	Yes	Lung, BALT	Wedge resection+ Rituximab	NED	70	No
6	53/F	Body	Positive	Yes	Rectum, MALT	Chemotherapy+RT	NED	74	No

Abbreviations: HP, *Helicobacter pylori*; MALT, marginal zone lymphoma of mucosa-associated lymphoid tissue; BALT, marginal zone lymphoma of bronchus-associated lymphoid tissue; DLBL, diffuse large B cell lymphoma; FU, follow up; NED, no evidence of disease; DRFS, distant relapse free interval.

**Table 4 cancers-13-00169-t004:** Univariate and multivariate analyses for relapse free survival (RFS).

		N	5-Yr RFS (%)	*p* ^(a)^	H.R (95% C.I)	*p* ^(b)^
Age	<53 years	72	93.2	0.3		
	≥53 years	73	91.7			
Sex	Male	80	91.5	0.9		
	Female	65	93.6			
Location of dominant lesion	Body	97	98.9	0.006	1.0	
	Others	48	81.5		5.85 (1.49–22.9)	0.011
H. pylori at diagnosis	Positive	61	94.8	0.5		
	Negative	84	90.2			
Antibiotics	Yes	87	94.4	0.7		
	No	58	89.5			
RT dose	≤30 Gy	104	96.0	0.9		
	>30 Gy	41	87.3			
Autoimmune disease	Yes	7	100.0	0.4		
	No	138	91.9			

^(a)^ Log-rank test for univariate analysis, ^(b)^ Cox-proportional hazard model for multivariate analysis. Abbreviations: RT, radiation therapy; RFS, relapse free survival; H.R, hazard ratio.

## Data Availability

The data presented in this study are available on request from the corresponding author. The data are not publicly available due to privacy and ethical restrictions.

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
