# Peer review of "Long-Term Clinical Outcome and Predictive Factors for Relapse after Radiation Therapy in 145 Patients with Stage I Gastric B-Cell Lymphoma of Mucosa-Associated Lymphoid Tissue Type"

_cancers, 2021, doi:10.3390/cancers13020169_

Round 1

Reviewer 1 Report

The study performed by Heerim Nam et al. retrospectively studied the clinical outcome of RT in 145 patients without a prior history of non-H.p eradiation treatment and demonstrated that RT would result in favorable clinical outcomes in patients with stage I gastric MALT lymphoma. There are a few comments:

  1. The major critique for this study is the novelty as this issue has been extensively investigated by previous studies. However, considering the large number of this cohort, it is still of value to be documented. 
  2. For H.p negative early-stage gastric MALT lymphoma, the most common strategies are RT and Rituximab (R). Do the authors have a control cohort of R treatment to compare the effectiveness, or is there any case in this cohort that had received R treatment concomitantly? If so, is there any evidence showing the benefit from R?

Reviewer 2 Report

This manuscript, original research type, written by Dr. Heerim Nam et al., and with the title of "Long-term clinical outcome and predictive factors for relapse after radiation therapy in 145 patients with stage I gastric B-cell lymphoma of mucosa-associated lymphoid tissue type" focuses on the effect of radiotherapy (RT) in patients with gastric extranodal marginal zone lymphoma (EMZL) of mucosa associated lymphoid tissue (MALT lymphoma). The manuscript is well written, contains enough detailed information, it is easy to read and the length is very appropriate as it allows to read it quickly without effort. In this research the authors studied a series of 145 patients with gastric MALT who were treated with RT and concluded that RT resulted in favorable clinical outcomes and that the tumor location could be a predictive factor for relapse. Before publishing the manuscript the authors could address the following minor comments: 1- I think that the authors should include the information of the lines 231-232 in the abstract because I think it is very relevant information ("Patients with gastric MALT lymphoma were referred for RT if they had H. pylori-negative 231 disease or failed to respond to 2 weeks of antibiotic therapy for H. pylori eradication"). 2- The authors could add a figure with the diagnosis and treatment algorithm that they have used, to help the reader to understand better the methods and results found in this series. 3- The authors do not mention about histopathological features of the cases, is there any pathological information that was relevant (e.g. the FISH analysis, tumor immune micro-environment, proliferation index, etc.). 4- In the Table 1 there are many abbreviations. If there is no problem of space or formatting, maybe the authors could simply write the complete words. 5- In case of H. pylori eradication failure, the authors performed RT. Is the period of 2 weeks the standard? 6- The authors could expand the discussion, with comparison to their treatment strategy to that shown in UpToDate (file attached in this report).
